# Ethidium Binding to *Salmonella enterica* ser. Typhimurium Cells and Salmon Sperm DNA

**DOI:** 10.3390/molecules26113386

**Published:** 2021-06-03

**Authors:** Sandra Sakalauskaitė, Valeryia Mikalayeva, Rimantas Daugelavičius

**Affiliations:** 1Department of Biochemistry, Faculty of Natural Sciences, Vytautas Magnus University, 44404 Kaunas, Lithuania; Sandra.Sakalauskaite@vdu.lt (S.S.); Valeryia.Mikalayeva@lsmuni.lt (V.M.); 2Institute of Cardiology, Lithuanian University of Health Sciences, 50162 Kaunas, Lithuania

**Keywords:** ethidium, tetraphenylphosphonium, multidrug resistance, outer membrane permeability, efflux inhibitor, phenylalanyl-arginyl-β-naphtylamide, Polymyxin B

## Abstract

Bacterial resistance to antibiotics due to increased efficiency of the efflux is a serious problem in clinics of infectious diseases. Knowledge of the factors affecting the activity of efflux pumps would help to find the solution. For this, fast and trustful methods for efflux analysis are needed. Here, we analyzed how the assay conditions affect the accumulation of efflux indicators ethidium (Et^+^) and tetraphenylphosphonium in *Salmonella enterica* ser. Typhimurium cells. An inhibitor phenylalanyl-arginyl-β-naphtylamide was applied to evaluate the input of RND family pumps into the total efflux. In parallel to spectrofluorimetric analysis, we used an electrochemical assessment of Et^+^ concentration. The results of our experiments indicated that Et^+^ fluorescence increases immediately after the penetration of this indicator into the cells. However, when cells bind a high amount of Et^+^, the intensity of the fluorescence reaches the saturation level and stops reacting to the accumulated amount of this indicator. For this reason, electrochemical measurements provide more trustful information about the efficiency of efflux when cells accumulate high amounts of Et^+^. Measurements of Et^+^ interaction with the purified DNA demonstrated that the affinity of this lipophilic cation to DNA depends on the medium composition. The capacity of DNA to bind Et^+^ considerably decreases in the presence of Mg^2+^, Polymyxin B or when DNA is incubated in high ionic strength media.

## 1. Introduction

Bacterial resistance to antimicrobials is a worldwide problem. The key part of the resistance is efflux pumps (EPs) [1]. AcrAB–TolC complex in *Salmonella enterica* ser. Typhimurium belongs to the resistance-nodulation-division (RND) family of efflux transporters. EPs of this family are the most important in Gram-negative bacteria because they extrude clinically relevant drugs [2]. It is necessary to understand the peculiarities of efflux to overcome this problem. Such knowledge would be essential for more efficient usage of the available antibacterials and for the discovery of new ones.

Lipophilic cations ethidium (Et^+^) and tetraphenylphosphonium (TPP^+^) are well-known EP substrates [3,4,5,6]. Transmembrane difference of electrical potential (∆ψ, negative inside) drives the accumulation of lipophilic cations in the bacterial cytosol. Intracellular components bind a low amount of TPP^+^ [7], and therefore, this cation can be used to estimate the ∆ψ in bacteria and mitochondria [8]. The specific feature of Et^+^ is the affinity of this cation to the double helix of nucleic acids. Intercalation of Et^+^ into the double-stranded DNA or RNA considerably increases the fluorescence of this compound [9]. Therefore, measurements of Et^+^ fluorescence is one of the most popular methods for assay of EP activity in bacteria [10,11,12,13]. The intensity of fluorescence correlates with the amount of Et^+^ bound to DNA, and the latter is proportional to the intracellular concentration of this indicator. Measurements of Et^+^ fluorescence is a convenient method for analysis of the competition between efflux substrates [14,15] because it is possible to use several efflux indicators in the same sample. However, Et^+^ fluorescence in wells of the plates proceeds slower than the electrochemically assayed binding of this indicator to bacteria [14]. Besides this, a gradual decrease in the fluorescence was observed in experiments with bacteria [13,14], although the cell-bound Et^+^ was not destructed [13].

Here, we investigated the fluorescence of Et^+^ in suspensions of *S.*
*typhimurium* cells at conditions close to ones used for the electrochemical monitoring of interaction of this indicator with the same bacteria. During the potentiometric analysis using the Et^+^ selective electrode, samples of the bacterial suspension were taken from the vessels for immediate assays of the fluorescence of this indicator. In parallel to the monitoring of Et^+^ interaction with the cells, at the same incubation conditions, the electrochemical measurements of TPP^+^ accumulation in bacteria were performed. An efflux inhibitor phenylalanyl-arginyl-β-naphtylamide (PAβN) was used to analyze the role of RND-family pumps in extrusion of the indicators from intact and EDTA- or Polymyxin B-permeabilized cells. Results of experiments indicated that fluorescence of Et^+^ reaches the saturation level when millimolar concentrations of this cation accumulate in the cytosol of *S.*
*typhimurium* cells. Measurements of Et^+^ interaction with the purified DNA showed that the ability of DNA to bind Et^+^ depends on the composition of the incubation medium. The presence of Mg^2+^ or Polymyxin B (PMB), or just increase in the ionic strength of the medium considerably decrease Et^+^ binding to DNA.

## 2. Results

### 2.1. Simultaneous Measurements of Et^+^ and TPP^+^ Interaction with S. typhimurium Cells

The results of our previous experiments indicated [14] that fluorescence of Et^+^ in *S.*
*typhimurium* suspensions correlates with the amount of this indicator bound to the cells. However, there were considerable differences in the kinetics and the amplitude of changes, registered by the spectrofluorimetric and the electrochemical techniques. This can be explained by the differences in cell incubation conditions during assays in vessels for the electrochemical measurements and in plate wells for the registration of Et^+^ fluorescence. To reduce the impact of incubation conditions, we combined electrochemical and spectrofluorimetric analyses: during the electrochemical monitoring of extracellular Et^+^ concentration, samples of *S.*
*typhimurium* suspension were taken to microtubes, and the fluorescence was immediately measured. In parallel, the electrochemical monitoring of TPP^+^ concentration in the suspensions of cells incubated at the same conditions was performed (Figure 1).

The addition of *S. typhimurium* cells to 100 mM Tris buffer had a very weak effect on Et^+^ fluorescence because of the outer membrane (OM) barrier and efflux pumps. The wild-type (wt) cells accumulated very low amounts of both lipophilic cations (Figure 1a,b), but concentrations of the indicators in the medium considerably decreased, and the fluorescence of Et^+^ increased when efflux pump mutant ΔtolC cells were added (Figure 1c,d). Chelator of the divalent cations EDTA permeabilized the OM and induced the accumulation of the indicator cations by the cells of both strains, strongly increasing the fluorescence of Et^+^ in bacterial suspensions. The obtained results revealed that there was no considerable difference between the accumulated amounts of the indicators in *S.*
*typhimurium* wt cells, while ΔtolC mutant cells before permeabilization accumulated twice more Et^+^ than TPP^+^ (8 and 3.6 nmol respectively). The total amounts of accumulated indicators in EDTA-permeabilized cells were rather similar (Figure 1).

The most widely used inhibitor of the RND family pumps PAβN increased the accumulation of Et^+^ and TPP^+^, and at the same time, stimulated Et^+^ fluorescence. The maximal level of fluorescence was reached when PAβN concentration in wt cell suspension was 128 µM, but only 32 µM were needed for the maximal accumulation and fluorescence in the case of ΔtolC mutant cells (see Figure 1b,d). At higher PAβN concentrations, the fluorescence of Et^+^ did not change, although the curves of electrochemical measurements indicated the leakage of indicators to the medium (see Figure 1d). It should be noted that Et^+^ leakage was electrochemically registered when the intensity of the fluorescence was rather stable and stayed at the maximum level (see Figure 1d).

Depending on the concentration, polycationic antibiotic PMB permeabilizes the OM and depolarizes the plasma membrane (PM) [16]. Results of the potentiometric measurements demonstrated that effects of PMB on Et^+^ accumulation in *S.*
*typhimurium* cells of both strains were dependent on the presence of PAβN: PMB was able to induce an additional accumulation of Et^+^ in the absence of PAβN (see Figure 1a,c), but the leakage of this indicator from cells was observed in the presence of the efflux inhibitor (see Figure 1b,d). In the absence of PAβN, the addition of PMB considerably increased the fluorescence of Et^+^ (see Figure 1a,c).

In the presence of PAβN, Et^+^ fluorescence was at the maximal level, and the addition of PMB had no effect on it (see Figure 1b,d). Correlation between the accumulated amount of Et^+^ and the intensity of fluorescence was lost when the amount of the cells bound to Et^+^ reached ~13 nmol (see Figure 1d). The fluorescence of Et^+^ did not change when the PAβN concentration in the medium increased from 32 to 128 µM, although the cell-accumulated amount of this cation decreased by 34% (see Figure 1d).

These results indicate that PMB induces an additional accumulation of Et^+^ when the distribution of this ion between cells and the incubation medium is not in equilibrium. Permeabilization of the OM by EDTA and inhibition of the efflux by PAβN led to the equilibrium distribution of TPP^+^ across the PM, enabling calculations of the ∆ψ. In the case of wt cells, the maximal level of ∆ψ, calculated according to the amount of accumulated TPP^+^ in the presence of 64 µM PAβN, was 209 ± 3 mV. In ΔtolC cells, the highest amount of TPP^+^ was accumulated in the presence of 32 µM PAβN, and the calculated ∆ψ at these conditions was 230 ± 1 mV. The binding of Et^+^ to DNA does not allow to assay the intracellular concentration of the unbound form of this indicator and, correspondingly, to calculate ∆ψ.

### 2.2. Et^+^ Binding to S. typhimurium Cells

According to Rodrigues and colleagues [15], the possibility to use Et^+^ as a fluorescent efflux indicator depends on the concentration of this ion in the medium. Higher concentrations of Et^+^, exceeding the capacity of the efflux pumps, are expected to result in increased accumulations, which, if sufficiently high, can result in ethidium reaching DNA where it can readily intercalate.

We decided to explore in more detail the dependence of the interaction of Et^+^ with *S. typhimurium* cells on the concentration of this indicator. Taking higher initial Et^+^ concentrations in the medium, we added intact or heat-killed cells and determined amounts of Et^+^ accumulated by the cells (see Figure 2). Additions of EDTA to Tris medium induced accumulation of Et^+^ by the cells, but the permeabilizing effect of this chelator was considerably weaker compared to the 3 μM indicator containing medium (see Figure 1a,b). PMB induced a considerably stronger accumulation, but the amount of Et^+^ bound was not high compared to the cells, preliminary treated with Tris/EDTA (see Figure 2b). In the absence of EDTA, PMB was more efficient, and the equilibrium distribution of Et^+^ across the cell envelope was achieved in two minutes after PMB addition (see Figure 2b). At lower concentrations, i.e., 24 μM, the PMB-induced accumulation of Et^+^ was not stable, and a release of this indicator followed the accumulation (see Figure 2b). Et^+^ was only released to the medium after the second PMB addition. The results of these experiments indicate that in high concentrations, there is no direct correlation between the cell-bound amount of Et^+^ and the initial concentration of the indicator in the medium.

Because of the heat-inactivated efflux and permeabilized OM, after heating envelope of the cells demonstrated a very weak barrier to Et^+^. In spite of the absence ∆ψ, accumulating cations in the cytosol, the heat-inactivated cells bound maximum amount of Et^+^ (190 nmol, when the initial concentration was 96 μM, see Figure 2c). The accumulated amount of Et^+^ in heat-inactivated cells was stable, did not changed during the incubation period. Starting the same initial concentration, the heat-inactivated cells were binding 24–50% more Et^+^ compared to EDTA- and/or PMB-permeabilized cells (compare Figure 2b,c). Changes in intracellular DNA arrangement during the heating could be the reason for the higher amount of Et^+^ bound and stronger fluorescence.

In order to get more information on the role of envelope barrier in Et^+^ binding to the cells, experiments with intact and Tris/EDTA-permeabilized cells, as well as purified salmon sperm DNA were performed. Cells or DNA were added to 100- or 400-mM Tris buffer containing 3 μM Et^+^. In both buffers, intact cells did not bind Et^+^, and slow accumulation of this indicator was observed only after PMB addition. The concentration of Et^+^ in the medium immediately decreased after the addition of Tris/EDTA-permeabilized cells, but after the fast initial accumulation, a slow release of the indicator was observed, more clearly expressed in 100 mM buffer (see Figure 3). PMB induced an additional release of the cell-bound Et^+^. In general, the cells in 400 mM buffer accumulated less Et^+^ compared to ones incubated in 100 mM Tris. The permeabilized cells in 400 mM buffer accumulated ~7 nmol of Et^+^, and ~1 nmol was released after PMB addition, but in 100 mM medium, these values were around 10 nmol and 3 nmol, correspondingly. In both media, after the addition of PMB, the intact cells accumulated higher amounts of Et^+^ compared to the preliminary permeabilized cells.

After the addition of DNA to the medium, a very fast decrease in the Et^+^ concentration to the stabile level was observed, and in 100 mM Tris, the bound amount of Et^+^ (11 nmol) was considerably higher than in 400 mM buffer (~6.5 nmol). In both media, PMB released a considerable amount of bound Et^+^, and the left amount of the indicator bound in 100 mM buffer was even a bit lower than in 400 mM (around 4.6 and 5.6 nmol, correspondingly. The amount of Et^+^ bound to DNA after PMB addition was very close to the amount left in Tris/EDTA-treated cells at the same conditions (see Figure 3).

### 2.3. Binding of Et^+^ to DNA in Solutions of Various Composition

To learn more about how Et^+^ binding to DNA depends on the composition of the medium, we extended the potentiometric analysis of this process. During the first experiment, we measured the amount of Et^+^ bound to a certain amount (100 μg) of DNA, increasing the concentration of this indicator in 100 mM Tris buffer. We elucidated that saturation level was reached at 60 µM and higher concentrations of this cation after binding of 50 nmol of Et^+^ to 100 µg of DNA (see Figure 4).

Continuing experiments with the purified DNA, we explored how changes in the medium composition could affect the binding of Et^+^ to DNA. In experiments with DNA solutions, the initial fluorescence of DNA-bound Et^+^ in 400 mM Tris buffer was ~20% lower compared to 100 mM Tris/HCl. During the monitoring period, the fluorescence gradually decreased, and after 25–30 min, it was ~20–30% lower compared to the initial level (see Figure 5a,b). The presence of TPP^+^ or PAβN in DNA solutions had a weak effect on Et^+^ fluorescence. However, in the presence of PMB or Mg^2+^ (also Ca^2+^, data not shown), the levels of fluorescence were considerably lower and very similar in both concentrations of Tris (see Figure 5a,b).

In our previous experiments with *S.*
*typhimurium* cells [14], the strongest decrease in Et^+^ fluorescence was observed after the addition of PMB to the suspension of *S.*
*typhimurium* cells in 400 mM Tris/HCl buffer. Electrochemical Et^+^ measurements showed that binding of this indicator to DNA is a fast process, and the amount bound is rather stable. In the medium with 1.2 μM Et^+^ (see Figure 5c), the initial intensity of the fluorescence correlated well with the amount of this cation bound to DNA (Figure 5a,b). In 400 mM Tris/HCl buffer, DNA bound to a lower amount of Et^+^ than in 100 mM Tris, and PMB or Mg^2+^ in the medium decreased the bound amount. Mg^2+^ or PMB in 100 mM Tris/HCl buffer induced a slight time-dependent release of bound Et^+^ (see Figure 5c). The addition of PMB to DNA solution in 100 mM Tris buffer at the end of the experiment immediately decreased the amount of bound Et^+^. However, the effect of PMB addition to a DNA solution in 400 mM Tris was weak, as well as the addition of this polycationic antibiotic to a DNA solution in 100 mM Tris, already containing Mg^2+^ or PMB (see Figure 5c). This demonstrates that ability of PMB to displace Et^+^ depends on the amount of this indicator bound.

However, the binding of Et^+^ to DNA was a partly reversible process when the initial concentration of this indicator was 60 µM, and a time-dependent release of the initially bound Et^+^ was observed (Figure 5d). In the presence of PMB in 400 mM buffer, the potential of the Et^+^-selective electrode decreased to values lower than the initial ones before the addition of DNA. These results suggest that some lipophilic cationic compounds were released from salmon sperm DNA as a result of the interaction with PMB.

## 3. Discussion

Antibiotics and other noxious to bacteria compounds are pumped out of the cells by energy-dependent transporters, mainly by proton motive force-driven efflux pumps [1]. The energy-dependent processes in bacteria are sensitive to the conditions of incubation. Different efficiencies and kinetics of the efflux from *S.*
*typhimurium* wt cells determined at various laboratories [4,12,14] can be explained by different methods of the assay used and, correspondingly, different cell incubation conditions and procedures used during the evaluation of the efflux.

The inhibition of MDR pumps is an attractive way to increase the efficiency of antibiotics. Evaluation of the antibiotic MIC values in the presence of efflux inhibitors is a direct but slow method: it takes 16 and more hours to get the result. Assay of the kinetics of the efflux of indicator compounds is a direct and fast method to determine the capabilities of the cells to extrude different xenobiotics, including antibiotics. It takes less than 0.5 h and provides a possibility to screen the compounds–candidates to efflux inhibitors.

Here, we potentiometrically and fluorimetrically analyzed the efflux in *S.*
*typhimurium* cells determining the bound amounts of Et^+^ and TPP^+^. In liquid cultures of Gram-negative bacteria, the electrochemical real-time monitoring of the efflux using Et^+^ and/or TPP^+^ as the indicators, PAβN as the efflux inhibitor, and PMB as the permeabilizer of the cell envelope is a convenient method. Bacterial suspensions are thermostated and constantly stirred during the experiments, reagents can be added, and the samples taken for complementary analyses without any interruption of the registration and any changes of the incubation conditions.

The proton motive force of bacteria in slightly alkaline media consists in most of ∆ψ [8]. Despite a high ∆ψ, *S.*
*typhimurium* wt cells with the intact OM in 100 mM Tris/HCl buffer accumulate neither TPP^+^ nor Et^+^ ions. However, ΔtolC mutant cells slowly bind these indicators in the absence of the OM permeabilizing compounds, but the presence of EDTA or PMB drastically increases rates of influx (see Figure 1). In the absence of TolC, AcrB or other energy-dependent components of the RND and ABC family, transporters extrude the lipophilic cations to the periplasm instead of the incubation medium, as wt cells do. More efficient accumulation of Et^+^ by ΔtolC cells before permeabilization of the OM can be explained by higher selectivity to TPP^+^ of EPs, expressed in the absence of TolC. At low concentrations blocking the activity of RND family pumps, PAβN increases the accumulation of indicator cations and stimulates Et^+^ fluorescence. The maximum accumulation of the indicators and the maximum fluorescence of Et^+^ in ΔtolC mutant cells was achieved at a lower PAβN concentration than for wt cells (see Figure 1b,d). This difference clearly indicates that for inhibition of the efflux, PAβN must cross the OM, and this crossing is more efficient in the case of mutant. The same tendency was observed also for ∆AcrB cells [17].

The altered extrusion of Et^+^ ions from cells, lacking the major EP complex MexAB–OprM, was observed in experiments with *Pseudomonas aeruginosa* cells [13,18]. Xu and colleagues observed that the intensity of the fluorescence of cells with an inactive MexAB-OprM pump decreased below the fluorescence in buffer solution when Et^+^ accumulation in the cells reached some “critical” values [18]. This was explained by the degradation of Et^+^ in MexAB-OprM mutant cells or the efflux through the assembly of unknown efflux transporters [18]. The authors were considering ethidium bromide as a non-dissociated neutral molecule, entering the viable bacterial cells by passive diffusion. The possibility to measure Et^+^ by ion-selective electrode indicates that ethidium bromide is dissociated, and near Nernstian behavior of the electrode shows that the degree of dissociation is close to 1. Babayan and Nikaido [13] extracted the cell-accumulated Et^+^ and presented evidence that this indicator is not destroyed by the cells after accumulation. It was concluded that self-quenching is the main reason for the decreased intensity of fluorescence. Our observations agree with the results from the Nikaido group that the accumulated Et^+^ is not degraded by the cells. The self-quenching could be due to the reduction of affinity of DNA to Et^+^ after the binding of a higher amount of this indicator. It is possible that PMB also more efficiently displaces this indicator from the binding sites when the high amount of Et^+^ is bound.

The results of experiments using pure DNA (see Figure 5) suggest that structural and/or functional changes inside the cells during the incubation period alter the DNA-bound amount of Et^+^. When binding equilibrium is reached in DNA solutions containing an excess of Et^+^, one Et^+^ cation is bound for every five nucleotides in DNA and one per ten nucleotides in RNA molecules [19]. There could be several reasons for fluorescence quenching. According to Hayashi and Harada [20], the intercalation of Et^+^ lengthens and unwinds DNA, and the isotherm of intercalation shows negative cooperativity between adjoining Et^+^ molecules. On the other hand, a concomitant drop in the intensity of fluorescence without a change in the amount of Et^+^ bound to DNA was observed with an increasing amount of the unbound Et^+^ [21]. In our experiments, a comparison of Et^+^ binding to purified DNA at low (1.2 μM) and high (60 μM) concentrations of this ion in the medium revealed that the release of accumulated Et^+^ to the incubation medium could be the main reason for the decrease in fluorescence at high concentrations of this ion (see Figure 5).

One milligram of the dry weight of *Escherichia coli* cells contains around 100 nmol of DNA base [22]. Considering that size *S.*
*typhimurium* chromosome (4857 kbp [23]) is like one of *E. coli* (4639 kbp [24]), in our experiments, 5 mL of *S.*
*typhimurium* suspension contained ~150 nmol DNA bases. The maximum intensity of the fluorescence was achieved when the amount of the bound Et^+^ was around 20 nmol. It means that the Et^+^/DNA ratio is less than 0.2, even when we consider also the RNA bases (about 20 nmol in 0.03 mg dry weight cells [22]).

We start to register the increase in fluorescence of Et^+^ simultaneously with the potentiometrically observed influx of this indicator into the cells. It is also indicated that primary binding occurs up to a bound-Et^+^-to-DNA-nucleotide ratio of 0.20–0.25, and if the Et^+^/DNA ratio is below 0.14, there are no changes in the spectrum [25]. However, we have not detected any threshold for an increase in Et^+^ fluorescence after this indicator binding to DNA. It is possible that the increase in Et^+^ fluorescence starts before it intercalates into DNA. Et^+^ binding to DNA is the ionic strength, as well as medium composition-dependent, and the presence of Mg^2+^ in the medium decreases the binding of Et^+^ to DNA and the fluorescence of this compound (see Figure 5). The decrease in Et^+^ fluorescence during the incubation with DNA was observed also in the absence of PMB or Mg^2+^ (see Figure 5 a,b). This could be because DNA–ethidium complexes become more accessible to water, which is a highly efficient fluorescence quencher [26], and the removal of water increases the fluorescence.

There is evidence of additional non-intercalative, less fluorescence-enhancing sites involving electrostatic binding to nucleic acids. The enhancement of Et^+^ fluorescence observed due to electrostatic binding to DNA or to hydrophobic solvents is attributed to a reduction in the excited state proton transfer rate [19]. However, this secondary binding occurs only at low ionic concentrations (i.e., 0.01 M) and when binding at the primary site is saturated. Low ionic strength conditions are not typical to the cytosol of viable bacteria, although it could happen after depolarization of the cells when Et^+^ fluorescence in bacterial suspensions is measured in low ionic strength media [11]. However, we must be careful in interpreting these results. Our results indicate (see Figure 2, also [14]) that heat-killed cells bind considerably more Et^+^ than intact cells, despite the very strong accumulation of the cations in the cytosol by membrane potential. These data are in agreement with [27] that fluorescence of metabolically inhibited cells never exceeded more than 50% of the dead cell values. Most probably, after cell death, the complex nature of bacterial chromosomes [28] is lost, and the relative amount of Et^+^ accessible double-stranded DNA considerably increases.

According to Rodrigues and colleagues [16], at concentrations lower than 3 μg/mL (7.5 μM), Et^+^ does not bind to DNA but already is a substrate of efflux pumps, and such concentrations should be used for studies of the efflux. Results of our experiments indicate that an increase in fluorescence is observed when Et^+^ concentration in the medium is lower, only 3 μM, and we have not found any concentration threshold for binding of this indicator to DNA. On the other hand, there are ideas that ethidium bromide becomes strongly fluorescent when it gets into the periplasm of Gram-negative bacteria or into the cytoplasm of Gram-positive ones [29].

When cells accumulate a high amount of Et^+^ and fluorescence reaches the maximum level, it stops correlating with the intracellular concentration of this cation, and Et^+^ loses the role of efflux indicator. At increased cell depolarizing concentrations, PAβN and PMB cause the release of accumulated Et^+^. The leakage of TPP^+^ indicates the total depolarization of the PM, but a considerable amount of Et^+^ remains inside the cells, most probably, because of the binding to nucleic acids. Our results indicate that the decrease in intracellular concentration of this cation not immediately leads to the decrease in the fluorescence. It looks that the release of bound Et^+^ is rather a slow process [14]. The fast release of Et^+^ after additions of PAβN and PMB agrees with [28] that if Et^+^ is not intercalated between nucleic bases of DNA, it is subject to extrusion or leakage after depolarization of the plasma membrane, as in our case. When it is intercalated, the binding constant is sufficiently high to keep Et^+^ from access to the efflux pump systems of the bacterium [19]. The dependence of Polymyxin B-induced Et^+^ binding to the cells on the presence of PAβN in the medium and the kinetics of Et^+^ binding/release indicate a complex nature of this process. Depending on the concentration used, PMB initially increases the permeability of the OM only and does not affect the PM of gram-negative cells [16]. At higher concentrations, PMB damages the PM and switches off energy-dependent processes, including efflux. When the cell suspension does not contain PAβN and Et^+^, fluorescence is not at its maximum, we see the additional Et^+^ accumulation after PMB addition (see Figure 1a,c). The permeabilization of the OM and switching off the efflux by PMB facilitates Et^+^ entry into the cytosol and binding to DNA, although depolarization of the PM repeals the ∆ψ-dependent Et^+^ accumulation-driving component. However, our previous experiments [14] demonstrated that such Et^+^ accumulation in *S.*
*typhimurium* wt cells is only temporary: PMB causes the leakage of Et^+^ after the equilibrium is reached. This means that decreasing fluorescence of Et^+^ after PMB addition is not only because of the self-quenching but also caused by displacement of Et^+^ by PMB at its binding sites. In the presence of EDTA, the effect of PMB on Et^+^ binding is weaker, most probably, because of the reduced binding of PMB to the OM due to EDTA-induced release of LPS [30] (compare curves in Figure 2a,b).

The presence of PAβN in the cell incubation medium considerably changes the run of events (see Figure 1b,d): depolarizing the PM and causing leakage of TPP^+^, PMB induces the release of more than 50% of accumulated Et^+^. How PAβN can so drastically change PMB effects on Et^+^ accumulation? Both compounds have a high affinity to LPS, but PMB added after PAβN displaces the latter at the binding sites (Sakalauskaite et al., in preparation). The increased concentration of the free PAβN, also subsequent added PMB, induces a release of accumulated Et^+^. The rather stable maximum intensity of the fluorescence in a situation when 50% of intracellular Et^+^ leaked out supports the idea that the release of DNA-intercalated Et^+^ is a slow process.

## 4. Materials and Methods

### 4.1. Bacteria Cultivation and Preparation for Experiments

*Salmonella enterica* ser. Typhimurium SL1344 cells of wild type (wt) and ΔtolC mutant strains were obtained from Prof. Séamus Fanning (Institute of Food and Health, University College Dublin, Ireland). Overnight cultures of these cells were grown in Luria-Bertani broth, containing 0.5% NaCl (Sigma-Aldrich, Munich, Germany), diluted 1:50 in fresh medium, and the incubation was continued until the OD_600_ reached 1. The cells were collected by centrifugation at 4 °C for 10 min at 3000× *g* (HeraeusTM MegafugeTM 16R, Thermo Scientific, Osterode am Harz, Germany). The pelleted cells were re-suspended in 100 mM Tris-hydroxyaminomethane (Tris)/HCl (Roth, Karlsruhe, Germany), pH 8.0, to obtain ~2 × 10^11^ cells/mL. The concentrated cell suspensions were kept on ice until used, but not longer than 4 h. For permeabilization of the outer membrane (OM) during the measurements, ethylene diamine tetraacetic acid (EDTA; Sharlau, Barcelona, Spain), pH 8.0, was added to the final concentration of 1 mM. To permeabilize the OM before measurements, the cells were at 37 °C 10 min incubated in 100 mmol/L Tris/HCl containing 10 mmol/L ethylene diamine tetraacetic acid (EDTA; Sharlau, Barcelona, Spain), pH 8.0, then pelleted and re-suspended as described above. Heat treatment of the cells was performed by incubating 1 mL of the concentrated suspension in a 1.5-mL Eppendorf tube for 10 min in a boiling water bath.

### 4.2. Potentiometric Measurements

TPP^+^ and Et^+^ concentrations in the incubation media were potentiometrically monitored using selective electrodes as described previously [14,17]. While assembling TPP^+^ and Et^+^-selective electrodes, the sensors were filled with 0.1 mM TPP^+^ chloride (Fluka, St. Gallen, Switzerland) or Et^+^ bromide (Acros Organics, Geel, Belgium) solutions in 100 mM NaCl and connected to internal Ag/AgCl half-cell electrodes. In between measurements, the sensors were stored dry at room temperature.

The thermostated and magnetically stirred glass vessels were filled with 5 mL of 100- or 400-mM Tris/HCl, pH 8.0, containing 0.1% glucose. After calibration of the electrodes, the concentrated cell suspension was added to obtain an OD_600_ of 1 or shared salmon sperm DNA stock solution was added to the final concentration of 20 mg/L. We used the electrode potential-amplifying system with an ultralow-input bias current operational amplifier AD549JH (Analog Devices, Norwood, MA, USA). The data acquisition system PowerLab 8/35 (AD Instruments, Oxford, UK) was used to connect the amplifying system to a computer. The agar salt bridges were used for indirect connection of the Ag/AgCl reference electrodes (Thermo Inc., Chelmsford, MA, USA; Orion model 9001) to cell suspensions or DNA solutions in the vessels. The measurements were performed simultaneously in 2–4 reaction vessels. The ∆ψ values were calculated as described previously [31], assuming that OD_600_ 1 corresponds to 8.3 × 10^8^ cells/mL, 2.3 × 10^9^ cells correspond to 1 mg of dry mass, and the intracellular water volume of *S. enterica* is 1.1 mL/g of dry mass. The representative sets of curves from 3–5 independent series of measurements are presented in figures.

### 4.3. Fluorescence Measurements

Single tube measurements were performed using microtubes with 75 µL of samples taken from vessels for electrochemical measurements. The intensity of Et^+^ fluorescence was measured by Modulus™ Single Tube Reader (Turner BioSystems, Inc., Sunnyvale, CA, USA) using a green filter set (excitation 525 nm, emission 580–640 nm).

For evaluation of Et^+^ interaction with sheared salmon sperm DNA (Eppendorf AG, Hamburg, Germany), stock solutions of PAβN hydrochloride, Polymyxin B (PMB) sulphate (7730 U of PMB base/mg; Sigma-Aldrich, Munich, Germany), TPP^+^ chloride, or MgCl_2_ (Roth, Karlsruhe, Germany) were added to the incubation buffer and mixed. Then salmon sperm DNA solution was added to the final concentration of 20 mg/L and Et^+^ to the corresponding concentration. The samples were mixed and transferred into 96-well flat-bottom black plates, 100 µL per well. The relative intensity of the fluorescence (excitation 535 nm, emission 612 nm) was monitored in the “TECAN GENios Pro™” (Männedorf, Switzerland) plate reader, thermostatting the plate at 37 °C. The plate was shaken 5 s before each registration point.

## 5. Conclusions

Electrochemical monitoring of the concentration of lipophilic indicators provides information on the kinetics of efflux and serves as an express method for studies of the efficiency of these transporters. Inhibiting the activity of efflux, PAβN, the most popular inhibitor of RND family pumps, increases the accumulation of Et^+^ and TPP^+^ in the cells and stimulates Et^+^ fluorescence. Lower concentrations of PAβN are needed to reach the maximal level of accumulation of lipophilic cations in the case of ΔtolC mutant than wt cells. Electrochemical measurements indicate that the main reason for decreasing Et^+^ fluorescence at high concentrations of this ion is the release of accumulated Et^+^ to the incubation medium. However, at the maximum level of Et^+^ fluorescence, PAβN, as well as PMB, cause the leakage of Et^+^ ions from the cytosol depolarizing their plasma membrane, but this leakage is not considerably affecting Et^+^ fluorescence. Et^+^ binding to DNA is medium ionic strength, as well as the composition-dependent, and the presence of PMB or Mg^2+^ in the medium decreases it. The intensity of fluorescence reaches the saturation level and stops reacting to the intracellular concentration of this indicator when cells accumulate a high amount of Et^+^.

## Figures and Tables

**Figure 1 molecules-26-03386-f001:**
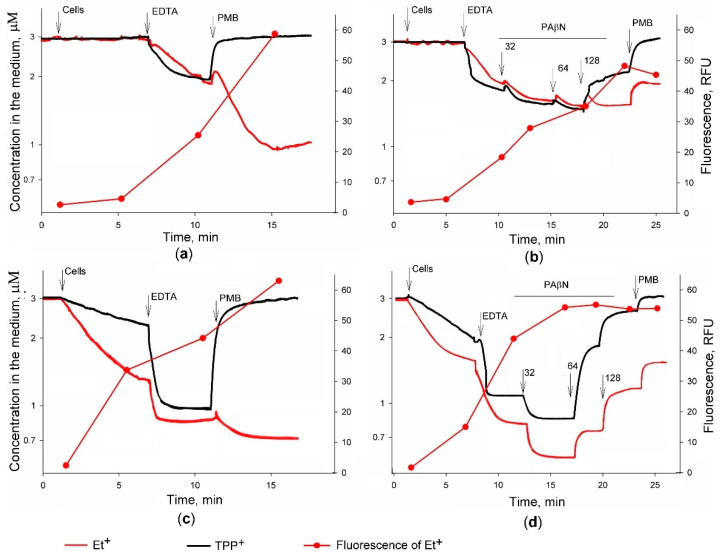
The accumulation of Et^+^ and TPP^+^ ions in *S.*
*typhimurium* SL1344 wt (**a**,**b**) and ∆tolC mutant (**c**,**d**) cells during simultaneous potentiometric and spectrofluorimetric assays. The measurements were performed at 37 °C in 100 mM Tris/HCl buffer, pH 8.0, containing 0.1% glucose. The concentrated cell suspensions were added to obtain OD_600_ of 1. EDTA was added to 1 mM and PMB to 50 mg/L. The final concentrations of PAβN (μM) are indicated in the figure (**b**,**d**). Samples of 75 μL each were taken from the vessels to microtubes just before the indicated additions, and intensities of the fluorescence were immediately measured.

**Figure 2 molecules-26-03386-f002:**
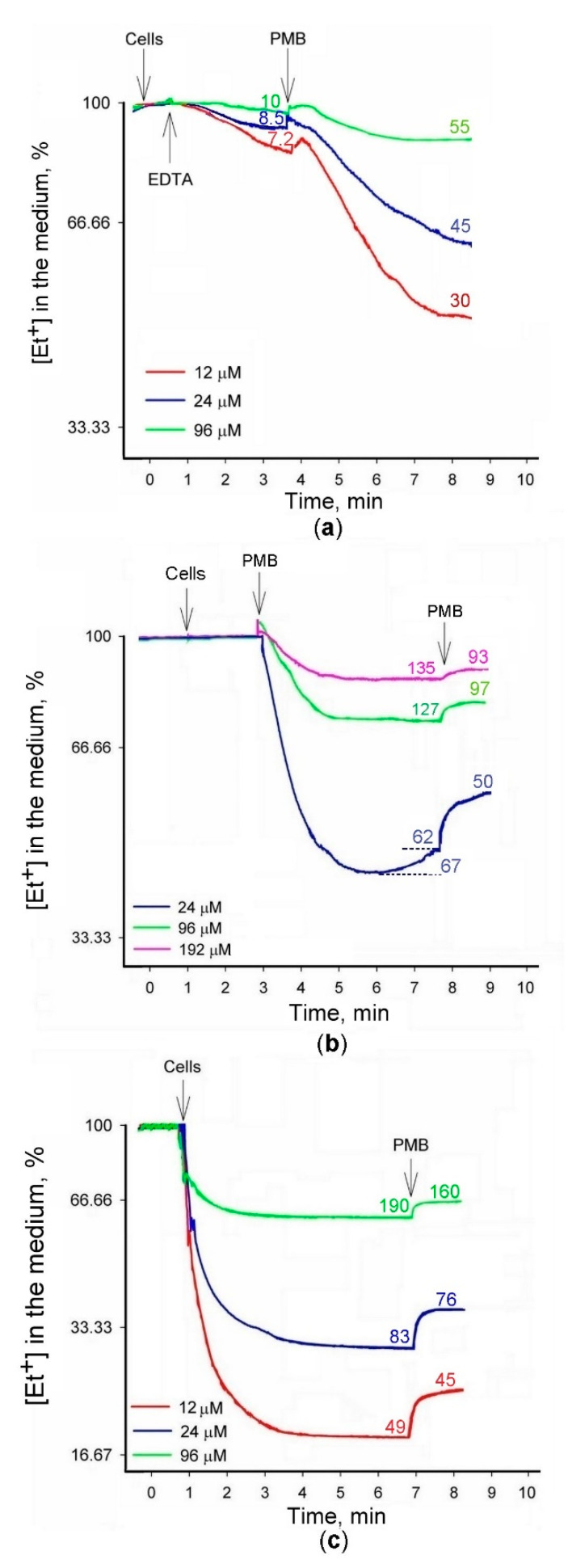
The binding of Et^+^ to intact and heat-inactivated *S.*
*typhimurium* SL1344 wt cells incubated with different concentrations of this cation. Concentrated intact (**a**,**b**) or heat-inactivated (**c**) cell suspensions were added to 100 mM Tris/HCl buffer, pH 8.0, containing 0.1% glucose, to obtain OD_600_ of 1. On Y-axes, Et^+^ concentrations are presented in % of the initial ones. The initial concentrations of Et^+^ are indicated in the figure. PMB was added to the final concentration of 50 mg/L (**a**,**c**) or 50 and 100 mg/L (**b**), EDTA to 1 mM. Numbers next to the curves indicate the amount (nmol) of Et^+^ bound to the cells. The experiment was performed at 37 °C.

**Figure 3 molecules-26-03386-f003:**
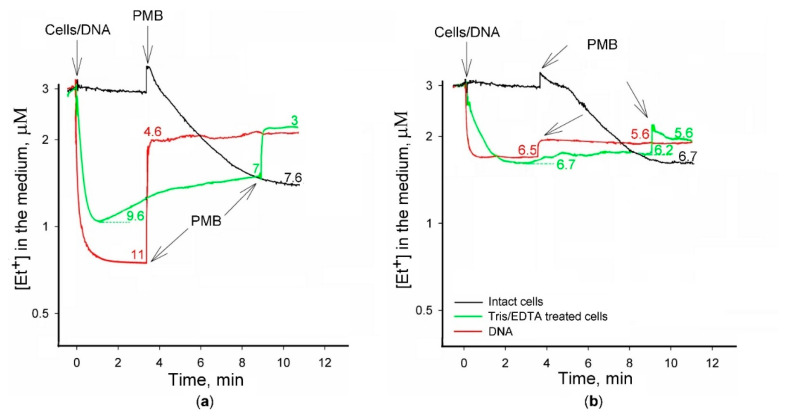
The binding of Et^+^ to intact or Tris/EDTA-permeabilized *S.*
*typhimurium* SL1344 wt cells and salmon sperm DNA. The experiments were performed at 37 °C in 100 mM (**a**) or 400 mM (**b**) Tris/HCl buffer, pH 8.0, containing 0.1% glucose. Concentrated suspensions of intact or Tris/EDTA-treated cells were added to obtain OD_600_ of 1. Red curves demonstrate the addition of 100 μg of salmon sperm DNA. PMB was added to the final concentration of 50 mg/L. Numbers next to the curves indicate the amount (nmol) of Et^+^ bound.

**Figure 4 molecules-26-03386-f004:**
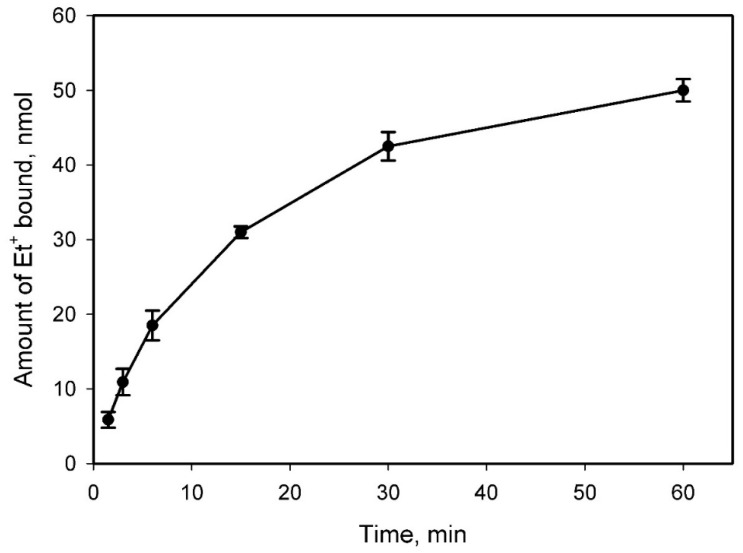
The dependence of the amount of Et^+^ bound to DNA on the concentration of this cation in the medium. In total, 100 μg of salmon sperm DNA was added to various concentrations of Et^+^ containing 100 mM Tris/HCl buffer, pH 8. The experiment was performed at 37 °C.

**Figure 5 molecules-26-03386-f005:**
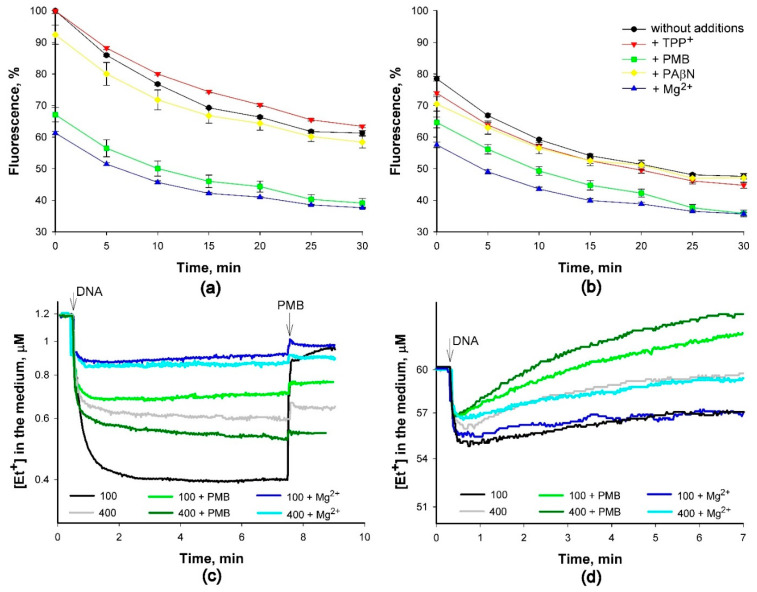
The influence of the medium composition on the interaction of Et^+^ with DNA. Experiments were performed at 37 °C in 100 mM (**a**,**c**,**d**) or 400 mM (**b**–**d**) Tris/HCl buffer, pH 8. For (**a**) and (**b**), 100% intensity corresponds to the initial level of Et+ fluorescence in DNA solution in 100 mM Tris/HCl buffer. The concentration of salmon sperm DNA was 20 mg/L. The initial Et^+^ concentration in (**a**–**c**) was 1.2 μM, in (**d**) 60 μM. Concentrations of PAβN, Mg^2+^, TPP^+^ and PMB were 250 μM, 10 mM, 125 μM and 100 mg/L, respectively. In c, at the end of the experiment concentration of PMB was increased to 200 mg/L. Label in (**b**) is valid for (**a**), also.

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
