# Peer review of "Ethidium Binding to Salmonella enterica ser. Typhimurium Cells and Salmon Sperm DNA"

_molecules, 2021, doi:10.3390/molecules26113386_

Round 1

Reviewer 1 Report

This work explores the kinetics of ethidium bromide binding to S. Typhimurium cells using fluorometric and potentiometric methods. Despite the relevance of studying and fighting bacterial resistance, the text has some deficiencies that need to be corrected for its better understanding and to properly assess its contribution to this field.

  • The text needs a profound grammar and language check. Because of this deficiency, several parts of the text are difficult to understand.
  • In Introduction, it is not clear why is important to describe the kinetics of Et+ efflux; the authors give an extensive explanation of the experiments done but did not remark the reason behind all the experiments. Moreover, would be useful to the reader to know some data (e.g., statistics, other efflux mechanisms, etc.) of  bacterial resistance to give proper attention to the subject and, thus, comprehend the relevance of this work.
  • In Results, it is not clear how the authors determined the amount of E+ since the data presented in Figures 1-3 is concentration of E+. Perhaps the data related to binding of Et+ to DNA in different conditions must come first to understand this data conversion.

Also, Figures' footnotes are rather confusing, not identifying clearly and briefly what are the conditions used to generate the data of each graph. For instance, in Figure 5, which are the graphs corresponding to 100 mM and 400 mM of TrisHCl? since the footnote (or perhaps the graph) is misleading.

In Figure 5, how can be explained  the increase of Et+ in the medium beyond the original concentration when PMB was added?

Is there any data related to the concentration of TPP? Some data is mentioned in the text, but there are no graphs or tables related to this analyte, which the text even mention that a selective electrode was built to determine its concentration.

  • In Discussion, the effects of heating and PMB on cell membrane permeability must be clearly stated to better correlate the data found in the experiments. Moreover, the comparative discussion of the data with one observed in MexAB-OprM must be better written because the current text is somewhat misleading.
  • The Conclusions seem a summary of what was observed, because the text does not emphasize the importance of determining the kinetics of the efflux of Et+ and its relevance in fighting bacterial resistance.

Author Response

I reviewer

This work explores the kinetics of ethidium bromide binding to S. Typhimurium cells using fluorometric and potentiometric methods. Despite the relevance of studying and fighting bacterial resistance, the text has some deficiencies that need to be corrected for its better understanding and to properly assess its contribution to this field.

  • The text needs a profound grammar and language check. Because of this deficiency, several parts of the text are difficult to understand.
  • The text was corrected. I hope, now it will be easier to understand.

  • In Introduction, it is not clear why is important to describe the kinetics of Et+ efflux;
  • Such paragraph was added to Discussion: Inhibition of MDR pumps is an attractive way to increase efficiency of antibiotics. Evaluation of the antibiotic MIC values in the presence of efflux inhibitors is a direct, but slow method: it takes 16 and more hours to get the result. Assay of the kinetics of the efflux of indicator compounds is a direct and fast method to determine the capabilities of the cells to extrude different xenobiotics, including antibiotics. It takes less than 0.5 h and provides a possibility to screen the compounds – candidates to efflux inhibitors.

  • the authors give an extensive explanation of the experiments done but did not remark the reason behind all the experiments.
  • Additional sentences were added to the text explaining the reasons behind the experiments.

  • Moreover, would be useful to the reader to know some data (e.g., statistics, other efflux mechanisms, etc.) of bacterial resistance to give proper attention to the subject and, thus, comprehend the relevance of this work.
  • As explained in the figure legends, typical experimental curves are presented in the figures, selected from three (or more) measurements. When two and more curves are presented in the figure, all of them are recorded with the same batch of cells.

Salmonella Typhimurium cell genome contains genes of at least nine efflux pumps. These experiments are focus on clinically the most relevant pumps of RND family, sensitive to the specific inhibitor PAbN. Some additional sentences were added to the text.

  • In Results, it is not clear how the authors determined the amount of E+ since the data presented in Figures 1-3 is concentration of E+. Perhaps the data related to binding of Et+ to DNA in different conditions must come first to understand this data conversion.
  • In Fig. 1, the medium initially contained 15 nmol of Et+ or TPP+ (volume 5 ml, concentration after calibration 3x10-6 M). After addition of cells, some amount of indicator ions was bound to the cells. Amount of free indicator left in the vessel = concentration of free (not bound to bacteria) indicator multiplied to the volume of the suspension. Using selective electrodes, we measure concentration of the free Et+, and volume of the suspension is known: 5,0 ml of the medium plus volume of the added bacterial suspension (~0.050 ml), plus volumes of all additions after the cells. Amount of the indicator bound to the cells = initial amount minus free amount.

The authors give an extensive explanation of the experiments done but did not remark the reason behind all the experiments I Also, Figures' footnotes are rather confusing, not identifying clearly and briefly what are the conditions used to generate the data of each graph. For instance, in Figure 5, which are the graphs corresponding to 100 mM and 400 mM of TrisHCl? since the footnote (or perhaps the graph) is misleading.

  • The Figure legends are written following the traditional format for description of curves of the electrochemical experiments. Legend for Fig. 5 is corrected.

In Figure 5, how can be explained the increase of Et+ in the medium beyond the original concentration when PMB was added?

  • This phenomenon is explained in Results (the last sentence of Results part).

Is there any data related to the concentration of TPP? Some data is mentioned in the text, but there are no graphs or tables related to this analyte, which the text even mention that a selective electrode was built to determine its concentration.

  • We have black curves in Fig. 1, reflecting interaction of TPP+ with the bacterial cells. Calculations of transmembrane difference of electrical potential are also based on data of TPP+ measurements.
  • In Discussion, the effects of heating and PMB on cell membrane permeability must be clearly stated to better correlate the data found in the experiments.
  • We have such sentence (lines 103-104) for PMB. One additional sentence is added to the text describing the effect of temperature.

  • Moreover, the comparative discussion of the data with one observed in MexAB-OprM must be better written because the current text is somewhat misleading.
  • Unfortunately, we do not understand, what is misleading in discussion of these results. In general, what was observed in experiments with P. aeruginosa, the same results we got with S. Typhimurium. Beside this, we showed, that the decrease of Et+ fluorescence leads to leakage of this cation from the cells.

  • The Conclusions seem a summary of what was observed, because the text does not emphasize the importance of determining the kinetics of the efflux of Et+ and its relevance in fighting bacterial resistance.
  • The Conclusions were corrected. A sentence on the importance of efflux kinetics measurements was added.

Reviewer 2 Report

The study by Sakalauskaite et al., titled Ethidium binding to Salmonella enterica ser. Typhimurium Cells and Salmon sperm DNA describes a clever way to monitor the function of efflux pumps using a combination of spectrofluorimetric and electrochemical detection of lipophilic cationic compounds, ethidium (Et+) and tetraphenylphosphonium (TPP+). Both compounds are known efflux pump substrates. There is a high demand for reliable assays that could assess the function of efflux pumps and provide the basis for screening for novel inhibitors. While the enthusiasm for this efflux pump assessment method is high, the study does need to address some major issues, including significant proofreading to correct numerous grammar issues.

Additionally, there is no clear rationale why each experiment was performed and what was learned from it. The rationale and conclusion for each figure have to be clearly stated.

What is not clear is why do PMB and PAbN decrease Et+ binding to DNA in a solution? This would indicate that changes mediated by these two compounds on the fluorescence readout are independent of their effect on efflux pumps which would change to the interpretation of results in Figure 1. Is it possible that PMB quenches fluorescence from Et+? One crucial experiment is to measure the fluorescence of Et+ in the presence of a different concentration of PMB (in the absence of cells or DNA).

Major comments:

-Can the authors demonstrate that the phenotype is dependent on tolC by complementation?

-How reproducible are the experiments in figure 1. Are the averages graphed or only a single run?

-why did the addition of PMB decrease dye-DNA binding (Fig 3A, Fig 5)?

Some examples of minor comments (there are more):

-y-axis of Fig 1c is not readable

-Some of the text in Figure 2 is not readable due to poor image quality and small font

-Figure 5 has double labels

-What do the authors mean by heat-inactivating cells? Do you mean heat-killed?

-There should be no space between a number and % sign (i.e., like 158, 193, etc.)

-line 41: Et+ has an affinity for nucleic acids, not necessarily double helix.

-certain words in the title are capitalized (i.e., Cells) while others are not (i.e., sperm). Please make it consistent
-line 125: where did the 34% decrease come from?

-line 42: the authors meant to say DNA and RNA, not RNR.

-line 70: the authors state that “…fluorescence of Et+ I S. Typhimurium suspension correlate (should be correlates) with the amount of this indicator bound to the cells.” – Shouldn’t the dye get internalized as opposed to being bound to cells?

-line 93: change the “most popular” to “most widely used” or something along these lines

-line 91: where did the 8 and 3.6 nmol values come from?

-line 115: remove underline from degrees centigrade – this applies to the whole paper

-Line 118: Please write out “Seventy-five microliters” at the beginning of the sentence. A sentence never starts with a numerical. The same applies in line 309.

-Line 123: Where did 13 nmol come from?

-Line 125: where did 34% come from? It may be easier for the reader to include the actual values in the figure legend.

-Line 131 and 133: where do the Dy values come from?

-Line 141: “Choosing higher….” Please re-write this sentence. It is not grammatically correct.

-Line 184: Stabile?

-Line 186: “than” not “then”

-Line 237: respectively, not “correspondingly”

-Line 247: The sentence is not clear. Please rewrite

-Line 250 and 251: change “registered” and “registration” to a different word

-Line 257: change “thermostated” to a different word

-Line 261: Rewrite the first sentence. It does not make sense.

-Line 279: remove the period after the bracket and move the ref. to the end of the sentence

-Line 284: Please rewrite this sentence so it’s more clear

-Line 287: evidence, not evidences

-Line 291: “destructed” may not be the best word. Perhaps inactivated or degraded

-Line 292: the authors discuss the possibility of PMB displacing Et+ from DNA. Please elaborate more on this. Also, Figures 3 and 5 suggest that PMB does interfere with Et+/DNA binding

-Line 293: displaces

-Line 295: Rewrite the sentence. It is not grammatically correct

-Line 301: Please elaborate more on the negative cooperativity. It’s not clear how does this relate to this study. Also, Hayashi and Harada used ethidium bromide.

-Line 309: The first sentence does not make sense

-Lines 309-315: What does that mean?

-Line 319: 0.14?

-Line: Why did the authors start to write EtBr?

-Lines 350, 351, etc. gram in Gram-negative/positive should not be capitalized. Please refer to CDC nomenclature: https://wwwnc.cdc.gov/eid/page/preferred-usage#:~:text=Gram%20should%20be%20capitalized%20and,used%20as%20a%20unit%20modifier.&text=Greek%20letters%20are%20preferred%20to,nonproprietary%20name%20uses%20the%20word.

Author Response

II reviewer

The study by Sakalauskaite et al., titled Ethidium binding to Salmonella enterica ser. Typhimurium Cells and Salmon sperm DNA describes a clever way to monitor the function of efflux pumps using a combination of spectrofluorimetric and electrochemical detection of lipophilic cationic compounds, ethidium (Et+) and tetraphenylphosphonium (TPP+). Both compounds are known efflux pump substrates. There is a high demand for reliable assays that could assess the function of efflux pumps and provide the basis for screening for novel inhibitors. While the enthusiasm for this efflux pump assessment method is high, the study does need to address some major issues, including significant proofreading to correct numerous grammar issues.

  • Many thanks to you for careful reading of our text and correcting mistakes. I hope, most of grammar issues are fixed. The text was rescanned using Grammarly and spelling programs.

Additionally, there is no clear rationale why each experiment was performed and what was learned from it. The rationale and conclusion for each figure have to be clearly stated. 

  • Additional explaining sentences were added to the text.

What is not clear is why do PMB and PAbN decrease Et+ binding to DNA in a solution? This would indicate that changes mediated by these two compounds on the fluorescence readout are independent of their effect on efflux pumps which would change to the interpretation of results in Figure 1.

  • Our results show, that PMB has triple effect on Et+ fluorescence in bacterial suspension. Initially, immediately after addition to the cell suspension, PMB increases permeability of the outer membrane. This promotes Et+ entry into the cells and accumulation in the cytosol because of However, because of “self-promoted uptake”, PMB interacts and depolarizes the plasma membrane. The third stage is a slow interaction of PMB with bacterial DNA. PMB (5+), being a cation, binds to polyanionic DNA and induces the release of bound Et+. This stage is much faster in experiments with pure DNA. The same process was observed with Mg(2+) or Ca(2+).

The situation with PAbN is different. Low concentrations of this compound inhibit efflux, high concentrations (how high depends on the outer membrane permeability) depolarize the plasma membrane. PMB releases the DNA bound Et+, but effect of PAbN on Et+ fluorescence is very weak (Fig. 5).

Is it possible that PMB quenches fluorescence from Et+? One crucial experiment is to measure the fluorescence of Et+ in the presence of a different concentration of PMB (in the absence of cells or DNA).

  • In the absence of DNA or cells, the fluorescence of Et+ is very weak and additions of PMB do not change it. At low concentrations PMB permeabilizes the outer membrane only, induces Et+ entry into the cells and increases fluorescence. PMB depolarizes both membranes, when high concentrations are added, initially increasing the fluorescence. However, the decrease of fluorescence and Et+ leakage starts after ~30 min [14]. In experiments with DNA, Et+ fluorescence decreasing effect of PMB is getting stronger in time (see fig. 5a and b and compare with effects of Mg2+ after 5 and 30 min).

Major comments:

-Can the authors demonstrate that the phenotype is dependent on tolC by complementation?

- We have not checked the phenotype by complementation. This strain is a gift of Prof. Seamus Fanning (University College Dublin), it is clearly different from wt cells.

-How reproducible are the experiments in figure 1. Are the averages graphed or only a single run?

All curves presented in Fig. 1 a-d are from the single experiments using the same batches of cells. Such experiments were repeated 3 (and more) times. Qualitatively the results were the same, but amounts of TPP+ and Et+ accumulated by the cells varied + 10%.

-why did the addition of PMB decrease dye-DNA binding (Fig 3A, Fig 5)?

- PMB increases the fluorescence, when it increases permeability of the bacterial envelope and promotes Et+ entry into the cells. PMB decreases dye-DNA binding, when PMB binds to DNA. Binding of PMB to DNA inside the cells is a slow process, but it is fast in the case of pure DNA.

 Some examples of minor comments (there are more):

-y-axis of Fig 1c is not readable

Corrected

-Some of the text in Figure 2 is not readable due to poor image quality and small font

Corrected

-Figure 5 has double labels

Corrected

-What do the authors mean by heat-inactivating cells? Do you mean heat-killed?

Yes, it means ”heat-killed cells”. Corrected.

-There should be no space between a number and % sign (i.e., like 158, 193, etc.)

Corrected

-line 41: Et+ has an affinity for nucleic acids, not necessarily double helix.

As a cation, Et+ has an affinity for nucleic acids. However, Et+ intercalates and the fluorescence increases only in case of dsDNA or dsRNA.

-certain words in the title are capitalized (i.e., Cells) while others are not (i.e., sperm). Please make it consistent

Corrected

-line 125: where did the 34% decrease come from?

- From comparison of cell-bound amounts of Et+, when the medium contains 32 and 128 mM of PAbN.

-line 42: the authors meant to say DNA and RNA, not RNR.

Corrected

-line 70: the authors state that “…fluorescence of Et+ I S. Typhimurium suspension correlate (should be correlates) with the amount of this indicator bound to the cells.” – Shouldn’t the dye get internalized as opposed to being bound to cells?

- To increase the fluorescence, Et+ should be internalized and bound to DNA. However, during the electrochemical measurements, we monitor concentration of the “free” Et+ and it does not matter, is Et+ bound to cell surface, or is it internalized. However, amount of Et+ bound to the cells surface is very low (Fig. 3, black curve, before addition of PMB).  

-line 93: change the “most popular” to “most widely used” or something along these lines

- Changed

-line 91: where did the 8 and 3.6 nmol values come from?

These values come from calculations of the amounts of Et+ bound to the cells.

-line 115: remove underline from degrees centigrade – this applies to the whole paper

Corrected.

-Line 118: Please write out “Seventy-five microliters” at the beginning of the sentence. A sentence never starts with a numerical. The same applies in line 309

Corrected.

-Line 123: Where did 13 nmol come from?

- From calculations of the amount of Et+ bound

-Line 125: where did 34% come from? It may be easier for the reader to include the actual values in the figure legend.

- Again, these numbers come from calculations of the amount of Et+ bound by the cells. We directly measure the concentration of Et+, and the amounts of Et+ (‘free” or bound) can be calculated from the curves.

-Line 131 and 133: where do the Dy values come from?

- From the calculations. We have a reference in Materials and Methods explaining, how to calculate Dy.

-Line 141: “Choosing higher….” Please re-write this sentence. It is not grammatically correct.

- Re-written

-Line 184: Stabile?

- Yes

-Line 186: “than” not “then”

Corrected.

-Line 237: respectively, not “correspondingly”

Corrected.

-Line 247: The sentence is not clear. Please rewrite.

Re-written.

-Line 250 and 251: change “registered” and “registration” to a different word

Changed.

-Line 257: change “thermostated” to a different word

Usually, we use this word in our papers. Could you, please, suggest better word?..

-Line 261: Rewrite the first sentence. It does not make sense.

Re-written.

-Line 279: remove the period after the bracket and move the ref. to the end of the sentence

-Corrected

-Line 284: Please rewrite this sentence so it’s more clear

Re-written

-Line 287: evidence, not evidences

Corrected

-Line 291: “destructed” may not be the best word. Perhaps inactivated or degraded.

Corrected to “degraded”. “Destructed” was the word used by authors of the cited paper.

-Line 292: the authors discuss the possibility of PMB displacing Et+ from DNA. Please elaborate more on this. Also, Figures 3 and 5 suggest that PMB does interfere with Et+/DNA binding.

- Additional sentence added  

-Line 293: displaces

Corrected

-Line 295: Rewrite the sentence. It is not grammatically correct.

Re-written

-Line 301: Please elaborate more on the negative cooperativity. It’s not clear how does this relate to this study. Also, Hayashi and Harada used ethidium bromide.

- In general, all researchers are using ethidium bromide in their experiments. However, many of them consider, that ethidium bromide is not dissociated to Et+ and Br- in solutions. However, our results show, that ethidium bromide is dissociated. We describe this in Discussion, lines 286-288.

-Line 309: The first sentence does not make sense. Lines 309-315: What does that mean?

 - We think that this paragraph is important. These calculations describe capacity of DNA to bind Et+, compare it to data from literature.

-Line 319: 0.14?

- Corrected

-Line: Why did the authors start to write EtBr?

- As well as investigators from other laboratories, we used Ethidium bromide in our experiments.

-Lines 350, 351, etc. gram in Gram-negative/positive should not be capitalized. Please refer to CDC nomenclature: https://wwwnc.cdc.gov/eid/page/preferred-usage#:~:text=Gram%20should%20be%20capitalized%20and,used%20as%20a%20unit%20modifier.&text=Greek%20letters%20are%20preferred%20to,nonproprietary%20name%20uses%20the%20word.

- Corrected. Thank you very much for very detail analysis of our text.

Round 2

Reviewer 1 Report

After the review period, the text shows a notable improvement, addressing the doubts expressed in the previous review, so I do not have any additional comments, except that the text must have a language check because there are still some errors.

Reviewer 2 Report

Thank you for addressing all of the comments. Please proofread the final version to correct additional grammar errors (especially the use of articles). I suggest checking the manuscript using Grammarly or similar programs.   

Just a couple of examples (there are more):

Line 165: arrangement, not arangement 

Line 246: should read: Antibiotics and other compounds that are noxious to bacteria are pumped...